# Transformation towards Risk-Sensitive Urban Development: A Systematic Review of the Issues and Challenges

**Ahmad Farhan Roslan** [1,2,*], **Terrence Fernando** [1], **Sara Biscaya** [1] **and Noralfishah Sulaiman** [2]

1   THINKLab, University of Salford, 7th Floor, Maxwell Building, Salford M5 4WT, UK;
    t.fernando@salford.ac.uk (T.F.); s.biscaya@salford.ac.uk (S.B.)
2   KANZU Research, Resilient Built Environment, Universiti Tun Hussein Onn Malaysia (UTHM),
    Parit Raja 86400, Malaysia; nora@uthm.edu.my
*   Correspondence: a.f.b.roslan@edu.salford.ac.uk

**Abstract:** Risk-sensitive urban development is an innovative planning approach that can transform the way cities are built in order to face the uncertainties that arise from climate-induced disaster risks. However, the potential to initiate such a transformative approach has not materialized because of the many underlying issues that need to be understood properly. Therefore, this study conducted a systematic review to gather empirical evidence on the issues and challenges in implementing risk-sensitive urban development. The study identified forty-six issues and challenges under seven key themes that need addressing in order to facilitate the desirable transition: trade-offs, governance, fragmentation and silos, capacity, design and development, data, and funding. The issues and challenges that exist under trade-offs for negotiating solutions for risk-sensitive urban development and the governance of multiple stakeholders were identified as the top two areas that need attention in facilitating the desirable transition. This study also revealed that important information, such as scientific information, hazard and risk information, temporal and spatial information, and critical local details are not being produced and shared between stakeholders in decision-making. A profound participatory process that involves all the stakeholders in the decision-making process was identified as the pathway to ensure equitable outcomes in risk-sensitive urban development.

**Keywords:** risk-sensitive urban development; disaster risk reduction; climate change adaptation; decision-making processes; collaborative planning approaches

## 1. Introduction

Cities are complex systems composed of interacting social and physical structures [1,2]. Rapid urbanization, technological advances, and the climate crisis are bringing new challenges in the shaping of these cities for a sustainable future [3]. Furthermore, the vision for modernizing conventional urban areas to smart cities requires a strategic planning approach that promotes the effective integration of physical, digital, and human systems in the built environment [4]. As more than 68% of the world's population is expected to live in urban areas by 2050 [5], cities are increasingly becoming vulnerable to various urban hazards, such as floods, heat waves, landslides, and droughts, induced as a result of large-scale development projects and climate change [6–8].

Although many scholars have undertaken much research and several initiatives to influence the creation of sustainable development through better city planning [9–13], very little research has explored collaborative planning approaches that consider disaster risk reduction (DRR) and climate change adaptation (CCA) in the decision-making process to build resilient cities [14–17]. Thus, little research has been undertaken on the collaborative planning approaches that can resist, absorb, accommodate, and recover from the effects of a hazard in a timely and efficient manner, through the preservation and restoration of the essential basic structures and functions of cities, as defined in UNISDR [18]. Building

resilience requires paying greater attention to improving the adaptive capacity (anticipative, absorptive, and restorative) [19,20], and the coping capacity [21,22], of a city with regard to the potential future hazards that can arise because of possible climate change scenarios (as presented in the Intergovernmental Panel on Climate change (IPCC) report. Since the adverse effect of hazards can propagate across domains because of the complex interconnections that exist among various subsystems (physical, social, ecological, economic), triggering a catastrophic disaster situation in a city, future urban development requires relevant stakeholders across all sectors to embark on a more collaborative effort that considers cities as complex systems, accepting risks as an integral element in overall city planning [12,23]. Such a collaborative effort requires a broader set of stakeholders that include federal and state governments, ministries, agencies, city officials, practitioners (e.g., urban planners, engineers, risk professionals), the private sector, NGOs, and citizens [13].

However, influencing a collaborative planning approach that considers climate risks is challenging because of the silo-based practices conducted by different stakeholders with multiple interests entrenched in current urban planning processes [13,24]. As a result, the UN Sustainable Development Goals (SDGs) that are aimed at addressing global concerns, such as reduced inequalities (SDG 10), sustainable cities and communities (SDG 11), climate action (SDG 13), life on land (SDG 15), and partnerships for the goals (SDG 17) [25], are hard to accomplish. Therefore, there is an urgent need to promote collaborative risk-sensitive urban development approaches to make cities and human settlements inclusive, safe, resilient, and sustainable [26].

Leck et al. [13] suggest that transition towards risk-sensitive urban development requires more inclusive governance, stronger networked collaboration, collective actions, locally accountable leadership, and improved risk data and monitoring. However, the underlying issues, such as the power between stakeholders [27], coupled with different priorities, perspectives, and time horizons [28], and the need for a shared vision among stakeholders [29], are recognized as key challenges for such a transition. Hence, there is a need to establish a deeper understanding of these challenges that detract from implementing risk-sensitive urban development, and to explore possible solutions for overcoming them.

Therefore, the purpose of this study is to conduct a systematic review to extensively diagnose the issues and challenges in implementing risk-sensitive urban development and to determine the potential solutions for overcoming these challenges. It is envisaged that the outcome of this study will provide a foundation for initiating future research that can facilitate a transition towards risk-sensitive urban development. This study adopts a systematic review approach to thoroughly retrieve, critically appraise, and synthesize the results of multiple papers that focus on risk-sensitive urban development. The following research question was chosen as the basis for conducting the systematic review in this paper: "What are the issues and challenges in implementing risk-sensitive urban development in the urban design process?"

The paper is organized into several sections: an introduction to the emergence of risk-sensitive urban development (Section 1); the systematic review process adopted to diagnose the issues and challenges (Section 2); the results from the descriptive analyses of the literature that present the identified key themes, issues, and challenges (Section 3); a discussion on the issues and challenges relating to risk-sensitive urban development (Section 4); the conclusions of this study (Section 5).

## 2. Method for Systematic Review

This section outlines the process adopted to perform a systematic review to identify the key issues and challenges in implementing risk-sensitive and transformative urban development. The systematic review minimizes bias by identifying, screening, and accessing the most relevant studies to the topic comprehensively. The structured review processes described by Koutsos et al. [30], and Tawfik et al. [31], was adapted in this study, and aligned with the basic steps provided in the PRISMA guidelines (Preferred Reporting

Items for Systematic Reviews and Meta-Analyses) [32]. Figure 1 illustrates the systematic review process used in this study that comprises seven stages.

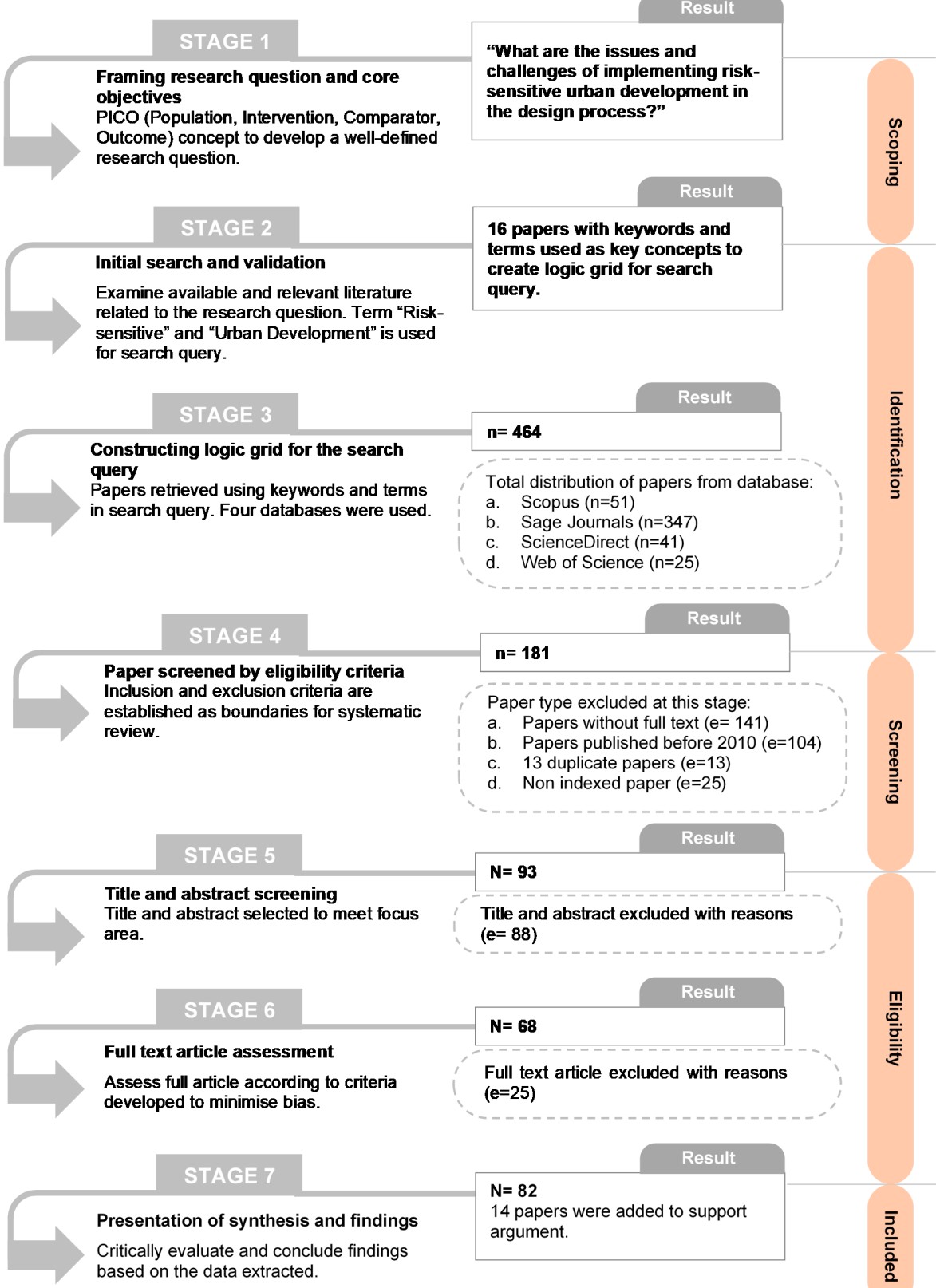

**Figure 1.** Overview of the systematic review process.

### 2.1. Stage 1: Framing the Research Question and Core Objectives

A well-defined research question for a systematic review should be formulated early to build the foundation for the search strategy and filters. The PICO (Population, Intervention, Comparison, Outcome) concept, as suggested by Aromataris & Riitano [33], was used in this study to frame the research question. PICO is an approach used for the systematic review and the meta-analysis of clinical trial studies [31]. However, this concept has been widely used in other fields, such as information technology [34], engineering and design management [35], and social science [36]. It assists researchers in retrieving relevant literature using appropriate terms in a searchable query during Stages 2 and 3. Following the PICO concept, the formulated research question was "What are the issues and challenges of implementing risk-sensitive urban development in the urban design process?". This research question can be broken down to map PICO as: P (population or problem): urban development; I (intervention or treatment of interest): risk-sensitive; C (comparator or control): urban design process; and O (outcome of interest): issues and challenges.

### 2.2. Stage 2: Initial Search and Validation

At this stage, the initial search needs to examine the available and relevant literature relating to the research question. This step is essential to ensure that relevant and adequate literature is chosen for conducting further analysis. According to Tawfik et al. [31], a simple search of the topic will provide more in-depth insight and gap identification to improve the research question or purpose. The terms "risk-sensitive" and "urban development" were used in the initial search query on three significant databases (Web of Science, Scopus, and ScienceDirect) which resulted in 16 papers. By studying the abstracts of the retrieved articles, a set of terms and keywords were then captured to create a substantial logic grid for the final search query.

### 2.3. Stage 3: Constructing a Logic Grid for the Search Query

A comprehensive search strategy should comprise both keywords, or free-text words, and index terms [33]. Table 1 shows the identified keywords for the search query within this topic. This step was used to assist in retrieving relevant and important papers pertinent to the search query. During this step, the use of keywords was utilized along with the appropriate selection of Boolean operators. The search was performed through the following four largest online databases: Scopus, SAGE Journals, ScienceDirect, and Web of Science. The availability of an extensive list of articles and journals covering multiple disciplines relating to urban studies was the rationale behind selecting these databases.

**Table 1.** Logic grid and identified keywords for the search query.

| Population | Intervention | Comparison | Outcome |
|---|---|---|---|
| Urban development *, Urban planning, Urban process, Urban design | Risk-sensitive, Risk sensitive, Risk sensitivity, Disaster risk *, Disaster reduction, Disaster risk reduction, Risk management, Risk reduction, Risk assessment, Risk evaluation, Risk based | Design process, Planning process *, Development process, Process design, Design method * | Issue *, Challenges *, Problem * |

Notes: * serve as the truncation or wildcard to retreive variation of term.

The literature retrieved using the search strings shown in Table 2 resulted in 464 papers. Different search strings were used on each database as the different databases handled the Boolean operators differently. Papers with citations generated from the search queries were exported to Bibtex format (*.bib file format) and recorded in the JabRef tool.

**Table 2.** The keywords and searching information (Boolean operators) for papers.

| Databases | Keywords and Searching Information | Number of Articles |
|---|---|---|
| Scopus | TITLE-ABS-KEY(("Urban development *" OR "Urban planning" OR "Urban process" OR "Urban design") AND ("Risk-sensitive" OR "Risk sensitive" OR "Risk sensitivity" OR "Disaster risk *" OR "Disaster reduction" OR "Disaster risk reduction" OR "Risk management" OR "Risk reduction" OR "Risk assessment" OR "Risk evaluation" OR "Risk based") AND ("Design process" OR "Planning process *" OR "Development process" OR "Process design" OR "Design method *") AND ("Issue *" OR "Challenges *" OR "Problem *")) | 51 |
| SAGE journals | Anywhere ("Urban development *" OR "Urban planning" OR "Urban process" OR "Urban design") Anywhere ("Risk-sensitive" OR "Risk sensitive" OR "Risk sensitivity" OR "Disaster risk *" OR "Disaster reduction" OR "Disaster risk reduction" OR "Risk management" OR "Risk reduction" OR "Risk assessment" OR "Risk evaluation" OR "Risk based") Anywhere ("Design process" OR "Planning process *" OR "Development process" OR "Process design" OR "Design method *") Anywhere ("Issue*" OR "Challenges *" OR "Problem *") | 347 |
| ScienceDirect (only 8 Boolean connectors) | Title, abstract, or author-specified keywords: (("Urban development" OR "Urban planning" OR "Urban process" OR "Urban design") AND ("Risk-sensitive" OR "Risk sensitive" OR "Disaster risk" OR "Disaster reduction" OR "Disaster risk reduction")) | 41 |
| Web of Science | TOPIC: (("Urban development *" OR "Urban planning" OR "Urban process" OR "Urban design")) AND TOPIC: (("Risk-sensitive" OR "Risk sensitive" OR "Risk sensitivity" OR "Disaster risk *" OR "Disaster reduction" OR "Disaster risk reduction" OR "Risk management" OR "Risk reduction" OR "Risk assessment" OR "Risk evaluation" OR "Risk based")) AND TOPIC: (("Design process" OR "Planning process *" OR "Development process" OR "Process design" OR "Design method *")) AND TOPIC: (("Issue*" OR "Challenges *" OR "Problem *")) | 25 |

Notes: * serve as the truncation or wildcard to retreive variation of term.

*2.4. Stage 4: Setting the Eligibility Criteria and Database Management*

The eligibility criteria used in this study are shown in Table 3. Using the feature in JabRef that can be utilized for filtering purposes, the following papers were removed: (a) 141 papers without full text; (b) 104 papers published before 2010; (c) 13 duplicate papers and 25 nonindexed papers. A total of 181 papers were finally retrieved and exported to an Excel file containing important information, such as the author's name, title, journal, publication year, URL link or DOI, and abstract for screening.

**Table 3.** Inclusion and exclusion criteria for systematic review.

| Inclusion Criteria | Exclusion Criteria |
|---|---|
| (a) Published between 2010 to 2020<br>(b) Peer-reviewed articles published in English<br>(c) Document Type: article, book, review, book chapter | (a) Grey literature (government or institution reports, newsletters, conference abstracts and proceedings, editorial material, conference papers)<br>(b) Nonindexed journal<br>(c) Articles without full text being available |

*2.5. Stage 5: Title and Abstract Screening*

The titles and abstracts of all the papers, filtered from the previous step, were thoroughly examined based on the focus of this study. The typology shown below describes how title and abstract screenings were examined. This process aimed to determine if the filtered studies satisfied the requirements of the focus area. In total, 93 relevant papers were selected for further review. The following key considerations were assessed for inclusion criteria:

(a)    Studies that investigate, describe, and assess issues and challenges relating to risk-sensitive urban development.
(b)    Studies that focus upon the topic of "urban studies", "planning and development", "disaster risk reduction", "environment", "resilience in the built environment", "urban design", "social studies", and "engineering".
(c)    Studies that assess the problems relating to disaster risk management, participatory design, community engagement, climate adaptation, adaptive governance, equitable resilience, and the urban design process.

*2.6. Stage 6: Full-Text Article Assessment*

A few key aspects were thoroughly examined to minimize the risk of bias. At this stage, a structured Excel sheet consisting of essential information was extracted and tabulated for assessment. The criteria included for this assessment in this study, adopted from Lunny et al. [37], and Whiting et al. [38], are presented in Table 4. Finally, 68 papers that were aligned with the research question of this study were thoroughly examined.

**Table 4.** Criteria included for full-text article assessment.

| Num. | Criteria for Assessment | Questions | Justification |
|---|---|---|---|
| 1. | The objective of the study | What is the objective of the study? | Only relevant articles related to the specific study area, including urban development, DRR, and CCA, are included in this study |
| 2. | Research method | What method is used in the study? | This study chooses articles that explain and provide a comprehensive methodological approach that can guarantee the rigor, quality, and value of the research |
| 3. | The population of the study | What is the problem or characteristics of the study? | The study includes articles that discuss problems faced by various stakeholders involved in integrating CCA and DRR into urban development |
| 4. | Intervention use in the study | How does the study wish to intervene or solve the problem? | The study includes intervention focusing on the type of transformation needed in implementing risk-sensitive urban development. The articles that address interventions (such as transformative approaches, including effective governance, balancing between development and disaster risk, new urban planning processes, collaboration between cross-sectors, decision-making processes, and capacity building) were selected in this study. |
| 5. | The outcome of the study | What is the possible outcome of the study? | The study includes articles that discuss the implications and suggestions for future improvement to achieve the desired outcome of the intervention. |

### 2.7. Stage 7: Presentation of Synthesis and Findings

The synthesis and findings of the 68 papers used in this systematic survey are presented in Section 3. However, a further 14 papers were later added in order to provide definition for the identified themes in the Discussion section (Section 4). These additional papers were also used in proposing potential solutions for the key challenges identified in this survey and suggesting recommendations for future research. A manual search, using theme keywords, was used to retrieve these papers.

## 3. Results

### 3.1. Descriptive Analysis of Literature

Figure 2 shows the classification of article publications relating to the field of risk-sensitive urban development over nine years (2010–2019). From early 2010 to 2014, the research trend relating to urban development increased as the United Nations Office for Disaster Risk Reduction (UNDRR) Making Cities Resilient Campaign (MCRC) launched in 2010. This campaign was part of the initiative to support the implementation of the Hyogo Framework for Action (HFA) at the local level until 2015, and currently continues with the implementation of the Sendai Framework for Disaster Risk Reduction (2015–2030) at all scales [39]. The results shown in Figure 2 indicate that the number of research publications began to decline between 2016 and 2019. One possible reason for the trend could be the growth of publications produced by international organizations (such as the UN Office for Disaster Risk Reduction (UNDRR), the Economic and Social Commission for Asia and the Pacific (ESCAP), the Global Facility for Disaster Reduction and Recovery (GFDRR), and the United Nations Development Programme (UNDP)), which are not scientific publications. While the researchers were aware of these nonscientific publications, it is important to note that this study only reviewed and analyzed high-quality scientific publications relating to the research questions, with limited information retrieved from nonscientific publications.

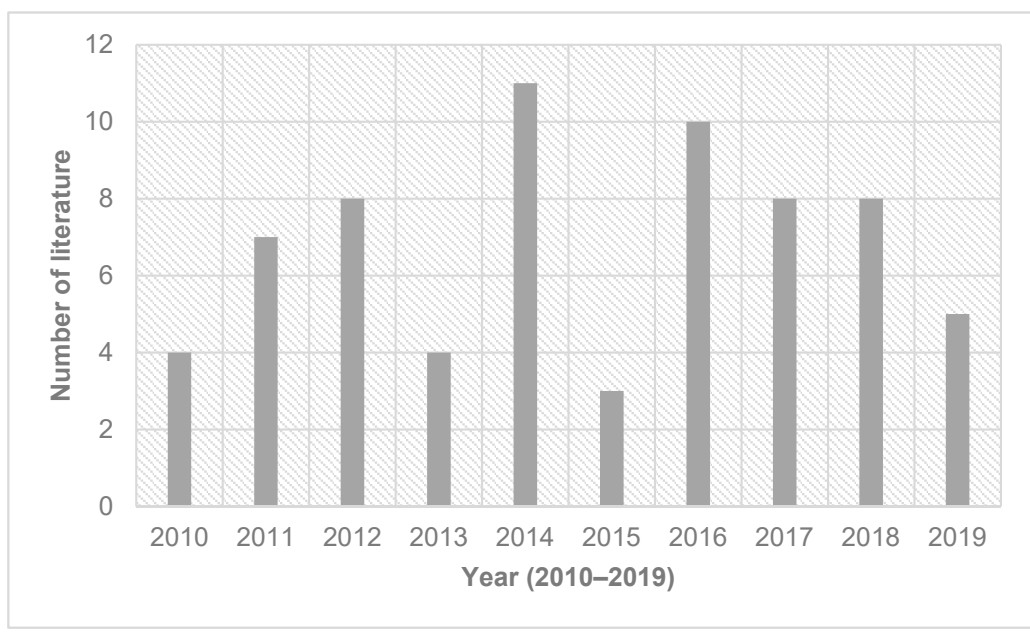

**Figure 2.** Annual distribution of articles.

Table 5 summarizes the number of studies conducted in various countries around the world. An analysis of the articles shows that the highest number of studies comes from the United States (12), the Philippines (9), South Africa (8), India (8), Indonesia (8) and Colombia (6). The research presented in these publications has been conducted in cities considered to be high-risk cities [40,41]. The United States has been the leading country in the field of disaster risk reduction, which can be mainly attributed to the

established 2011–2014 strategic plan by the Federal Emergency Management Agency (FEMA) in strengthening the nation's resilience to withstand catastrophic disasters as its main priority [42,43].

**Table 5.** Number of studies conducted by the country.

| Country | Number of Studies Conducted | Country | Number of Studies Conducted |
|---|---|---|---|
| United States | 12 | Denmark | 2 |
| Philippines | 9 | Lebanon | 2 |
| South Africa | 8 | Switzerland | 1 |
| India | 8 | Venezuela | 1 |
| Indonesia | 8 | Uganda | 1 |
| Colombia | 6 | Ethiopia | 1 |
| Sri Lanka | 5 | Sudan | 1 |
| Nicaragua | 5 | New Zealand | 1 |
| Australia | 4 | Japan | 1 |
| Chile | 4 | Canada | 1 |
| Thailand | 3 | Malawi | 1 |
| Argentina | 3 | Austria | 1 |
| United Kingdom | 3 | Cameroon | 1 |
| Kenya | 3 | Pakistan | 1 |
| China | 3 | Sweden | 1 |
| Italy | 3 | Peru | 1 |
| Tanzania | 2 | Costa Rica | 1 |
| Mexico | 2 | Bangladesh | 1 |
| Turkey | 2 | Bhutan | 1 |
| Netherlands | 2 | Portugal | 1 |
| Ecuador | 2 | Algeria | 1 |
| Germany | 2 | Solomon Islands | 1 |
| Vietnam | 2 | Vanuatu | 1 |
| Nigeria | 2 | Belgium | 1 |
| El Salvador | 2 | Brazil | 1 |
| Ghana | 2 | Spain | 1 |
| Nepal | 2 | Iran | 1 |

There is a wide distribution of the selected literature in a broad field of research and disciplines in the risk-sensitive urban development area. *The Journal of Environment and Urbanization* (19 papers) encompasses the largest number of research articles, followed by *The International Journal of Disaster Resilience in the Built Environment*, and *The Journal of Planning Education and Research* (eight each, respectively). The articles selected in this study are from highly indexed journals, covering broad areas, such as urbanization, planning, disaster and climate change, policy, and social sciences (see Table 6).

**Table 6.** Distribution of the selected literature and impact factor.

| Journal | Num. of Paper | Index (h-Index) | SJR Impact Factor 2018 |
|---|---|---|---|
| Environment & Urbanization | 19 | 62 | 1.44 |
| International Journal of Disaster Resilience in the Built Environment | 8 | 13 | 0.92 |
| Journal of Planning Education and Research | 8 | 63 | 1.04 |
| International Journal of Disaster Risk Reduction | 6 | 28 | 1.35 |
| Environment and Urbanization ASIA | 3 | 11 | 0.32 |
| Current Opinion in Environmental Sustainability | 2 | 69 | 1.98 |
| Environment and Planning A | 2 | 112 | 1.55 |
| Social Studies of Science | 1 | 78 | 1.16 |

**Table 6.** *Cont.*

| Journal | Num. of Paper | Index (h-Index) | SJR Impact Factor 2018 |
|---|---|---|---|
| Progress in Physical Geography | 1 | 92 | 1.38 |
| Journal of Planning Literature | 1 | 49 | 1.16 |
| Global Environmental Change | 1 | 147 | 4.38 |
| Science of the Total Environment | 1 | 205 | 1.54 |
| International Area Studies Review | 1 | 9 | 0.19 |
| Climatic Change | 1 | 162 | 1.64 |
| Environmental Science and Policy | 1 | 95 | 1.92 |
| Land Use Policy | 1 | 93 | 1.41 |
| Urban Climate | 1 | 28 | 0.89 |
| Public Works Management & Policy | 1 | 21 | 0.34 |
| Urban Studies | 1 | 131 | 2.12 |
| Smart and Sustainable Built Environment | 1 | 10 | 0.27 |
| Transportation Research Record: Journal of the Transportation Research Board | 1 | 94 | 0.54 |
| Environmental Science & Policy | 1 | 95 | 1.92 |
| Journal of Flood Risk Management | 1 | 28 | 1.08 |
| American Review of Public Administration | 1 | 47 | 2.08 |
| WIT Transactions on Ecology and The Environment | 1 | 19 | 0.13 |
| International Journal of Health Services | 1 | 52 | 0.69 |
| International Journal of River Basin Management | 1 | 31 | 0.42 |

### 3.2. Key Themes, Issues and Challenges Analysis

This study employed a thematic analysis process to capture and analyze textual data to answer the research questions [44,45]. The data was analyzed using computer-assisted qualitative data analysis software (CQDAS), namely, ATLAS.ti. The thematic analysis enables researchers to identify recurring patterns and themes with respect to the research question [46]. By adopting the thematic analysis process reported in Friese et al. [47], this study identified seven key themes: trade-offs, governance, fragmentation and silo, capacity, design and development, data, and funding. Among the seven key themes identified in this study, the theme of trade-offs was observed to be the most often discussed theme (23%), followed by governance (19%), fragmentation and silo (16%), capacity (13%), design and development (12%), data (10%), and funding (7%).

Forty-six issues and challenges were identified under these seven key themes, as illustrated in Figure 3. This study's analysis shows that the production of information (6.3%) is one of the greatest challenges faced in this area. Issues associated with information have been highlighted as the major cause that undermines effective decision-making processes. The second-highest challenge is the inadequate skills and technical capacity (5.7%) of the relevant stakeholders (such as local level institutions that impede efforts to reduce disaster and climate risk). Several key challenges were found under the trade-offs theme, such as politics and leadership (4.8%), a complex risk evaluation process (4.7%), a lack of understanding of adaptation needs and priorities (4.6%), and equity and equality (4.3%). The review indicates that issues, such as inadequate design consideration (4.0%), power relations and inappropriate development (3.8%), weak coordination (3.5%), and weak governance (2.9%), are also disrupting the implementation of risk-sensitive urban design.

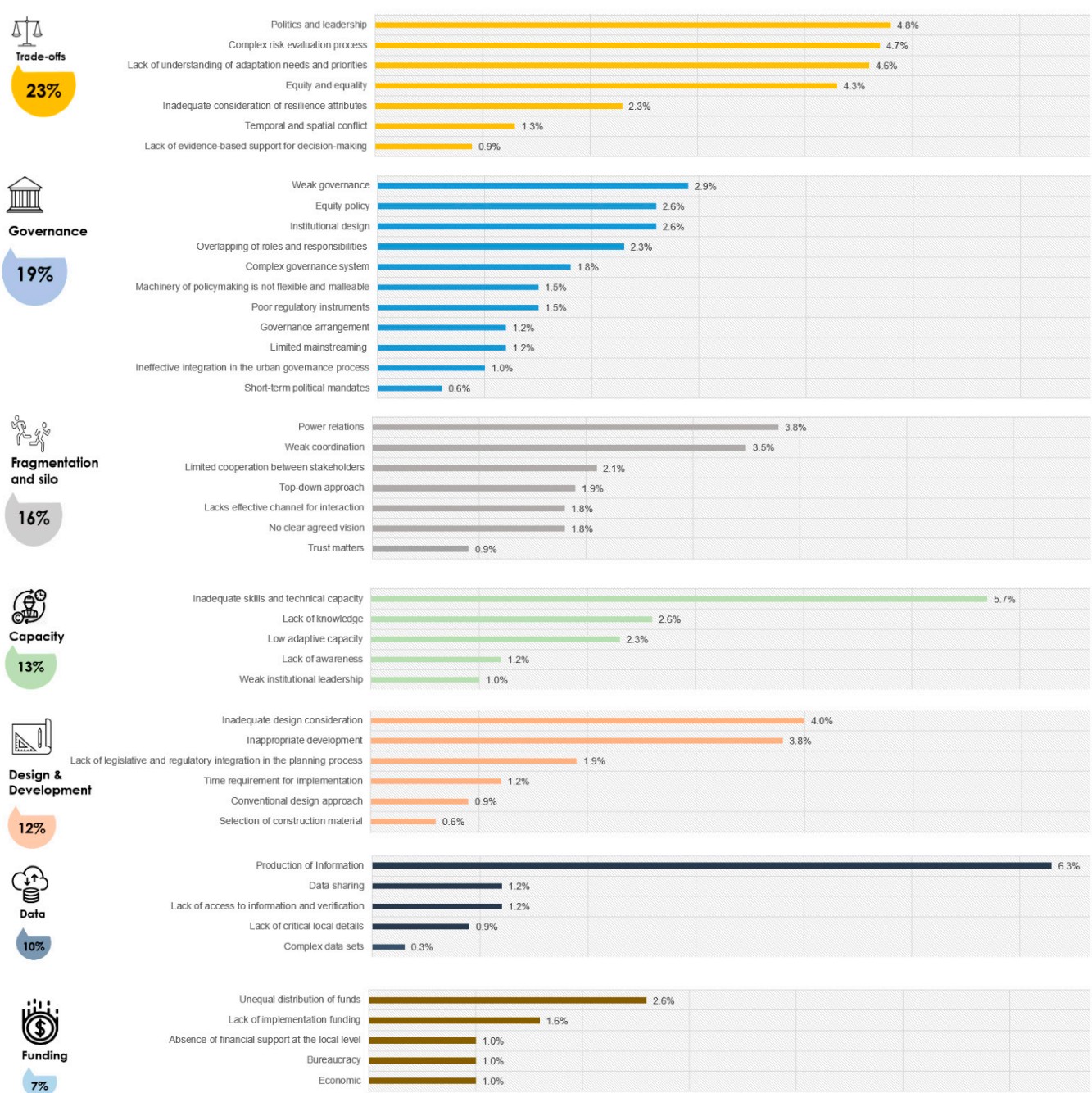

**Figure 3.** Issues and challenges within critical themes in implementing risk-sensitive urban development.

## 4. Discussion: Issues and Challenges to Risk-Sensitive Urban Development

The findings of the systematic review presented in this paper have been useful to draw out the key issues and challenges that need addressing in order to influence a transition towards risk-sensitive urban development. The themes generated have emerged from a rigorous and systematic coding process conducted by the authors. This section presents a discussion on the key themes identified in promoting risk-sensitive urban development. It specifically focuses on the need for open and transparent trade-offs in decision-making, strong governance, better institutional links that combat fragmentation and silo-based working practices, integrated design and development processes, capacity building, and data-driven decision-making and investment.

### 4.1. Trade-Offs

Table 7 shows issues and challenges under the trade-offs theme.

**Table 7.** Issues and challenges under the trade-offs theme.

| Issues and Challenges | Reference | Frequency (%) |
|---|---|---|
| Politics and leadership | [48–66] | 4.8 |
| Complex risk evaluation process | [48,50–52,54,55,57,58,60–62,67–78] | 4.7 |
| Lack of understanding of adaptation needs and priorities | [17,49,50,54,55,57,61,67–69,72,78–90] | 4.6 |
| Equity and equality | [50,53,55,68,69,72,74–78,80,82,87,88,90–94] | 4.3 |
| Inadequate consideration of resilience attributes | [54–57,59,69,72,80,81,89,90,95–97] | 2.3 |
| Temporal and spatial conflict | [56,59,74,76,83,89,90,98,99] | 1.3 |
| Lack of evidence-based support for decision-making | [55,62,63,65,94,100] | 0.9 |

Trade-offs are defined as complex situations in which a decision needs to be made between two desired goals as they cannot be achieved simultaneously [24]. Conflicts may arise, for example, if one group is interested in solutions that may exacerbate disaster risks and climate change while others are interested in reducing disaster risks and climate change. Conflicts of interest between public needs and political priorities [65] are among the most controversial issues that require solutions. In this context, concerns have been raised regarding the use of political power [66] to protect political and business interests [63]. The trade-off's dilemma in such situations requires transparency in decision-making, informed by scientific evidence. Ayson et al. [63] report how information is not being used to understand and analyze the associated trade-offs in relation to community needs, resource management, and the elements at risk. This review highlights the lack of an evidence-based approach to decision-making at present [53,54,56]. Without extensive evidence being available, patronage politics may come into the scene, resulting in decisions that favor the interests of a few [63].

Evidence-based decision-making in relation to climate and disaster risk reduction requires prioritizing actions based on a thorough risk evaluation. However, the nature of the risk evaluation process presents a multitude of challenges due to many factors, such as the complex relationships between the hazard and the elements at risk [76], difficulties in assessing the impact of risks during the planning process [60], and the difficulties in interpreting the information on hazards and risks by the stakeholders in land-use planning [95]. Furthermore, the temporal conflicts (the tension between short-term and long-term vision), and spatial conflicts (the tension between urban growth and land resource use) opens up another dimension in trade-offs [90]. In addition, key attributes for making cities resilient are, generally, not adequately considered in the planning process [55,66,90]. Trade-offs for enhancing the key attributes of resilience [101] rarely occur because of a lack of emphasis on city-resilience thinking during urban planning processes [102]. In this context, in-depth vulnerability analyses are rarely embedded in the planning process, with little attention given to the issues of risk reduction and adaptation to climate change [80,95,97]. The vulnerability analysis provides insight into the risks that exist in cities in many dimensions, such as the physical, social, environmental, economic, governmental, and health dimensions, guiding stakeholders to focus on the most critical areas that are considered highly vulnerable.

Risk evaluation is poorly understood from the community perspective because of limited community engagement [58,75,77]. Despite the fact that vulnerable communities are the most affected by hazards, little attention has been given to involving them in the planning process to achieve risk-responsive developments that are equitable. The case study evaluations conducted in [77,94] have shown that decision-making processes tend to be dominated by certain groups who have personal or business interests, those who are wealthier and educated, and those who have the time and resources to participate consistently. As a result, the marginalized groups are often overlooked as they are not strongly represented in community activities or political processes [67]. In this aspect,

trade-offs between equity and equality (distribution of risks, benefits and losses) need to emerge in the decision-making process [103]. All these trade-offs need to be addressed through open and transparent discussions through participatory planning that challenges existing systems, processes, social values, institutions, and technical practices. Participatory planning approaches have the potential to empower a wide range of stakeholders and help participants to understand the multiple perspectives of a complex problem, promoting systems thinking, improving stakeholder relationships, and promoting social learning to achieve better outcomes [104,105].

*4.2. Governance*

Table 8 shows issues and challenges under the governance theme.

**Table 8.** Issues and challenges under the governance theme.

| Issues and Challenges | Reference | Frequency (%) |
|---|---|---|
| Weak governance | [50,54,55,57,62,67,69,72,81,87,88,92,106,107] | 2.9 |
| Equity policy | [65,66,70,78,80,82,94,107–109] | 2.6 |
| Institutional design | [49,55,57,59,60,67,80,85,88,90,110–113] | 2.6 |
| Overlapping of roles and responsibilities | [49,53,54,56,61,62,66,68,76,95,110,111,114] | 2.3 |
| Complex governance system | [55,59,66,82,88,97,100,109,115] | 1.8 |
| Machinery of policymaking is not flexible and malleable | [54,61,65,66,71,99,110,111] | 1.5 |
| Poor regulatory instruments | [52,53,57,60,61,71,72,81] | 1.5 |
| Governance arrangement | [49,51,66,80,82,86,114] | 1.2 |
| Limited mainstreaming | [57,59,68,71,80,92] | 1.2 |
| Ineffective integration in the urban governance process | [51,53,57,82,109,114] | 1.0 |
| Short-term political mandates | [52,56,68,116] | 0.6 |

As described by Forino et al. [14], governance is the means of interaction among networks of actors, sometimes with conflicting objectives, and the instruments chosen to solve societal problems that will create societal opportunities in a particular area. In the urban climate context, governance can be explained as the process in which public, private, and civil societies and institutions articulate climate goals, with multiple stakeholders involved in decision-making through various processes at multiple scales of social organization [117]. At the core of this aspect lies the importance of better collaboration between the relevant stakeholders to effectively link DRR and CCA.

The shortcomings under this theme are mostly regarded as being the result of weak governance. Within this context, inabilities to adopt a participatory approach in complex urban governance prevail because of several reasons (including a lack of integrated planning and knowledge, and fragmented administrative systems and responsibilities in areas such as urban planning policy, disaster risk management, and climate change adaptation [72]). The prevalence of technocratic organizational structures [65], weak institutional coordination between agencies [100,118], complex federal bureaucracy [64], and top-down approaches [63,90,94] were identified as part of the institutional design issues that hinder participatory approaches. In this regard, policy development largely fails to prioritize actions that are critical and consistent in reducing risk for vulnerable populations [24]. According to Berke et al. [78], multiple groups may create plans to pursue their interests, resulting in a fragmented network of plans that are poorly coordinated, which could potentially conflict with each other and, subsequently, increase urban risks. As mentioned in [106], a governance dilemma can exist where subordinate governing bodies are unwilling to support policy developed by the higher level of government, creating a significant obstacle to implementing a collective approach to risk-sensitive urban development. In [78], Berke et al. suggest that one of the reasons for such unwillingness by the subordinate governing bodies to support policies is due to the inability of the higher-level government bodies to create equity policies that support risk reduction for vulnerable communities in their local context. It can be observed that cities tend to adopt climate

policies and hazard mitigation plans or implement strategies that are being developed at the national level, with limited engagement at the local level [94,107].

Although the participatory approach has been recognized as important, it requires further research, as multiple stakeholders with compartmentalized policies and practices need to be brought together to promote cross-sectoral collaboration within a complex governance system [55]. The involvement of many institutions with overlapping roles and responsibilities may lead to legal uncertainties and conflicts [62,114]. Additionally, each institution, with its own codes of practices, legislation, and policies, leads to poor regulatory instruments with competing policy agendas, lack of participation in plan creation, and weak coordination within urban institutional settings [60]. Hence, dealing with issues relating to climate and disaster impacts needs to be supported by an integrated and cross-cutting policy approach.

The integration of CCA and DRR measures into existing policies is difficult to achieve because of limited mainstreaming efforts [68,89]. According to Reckien et al. [119], and Uittenbroek et al. [120], mainstreaming (within the climate change literature) is defined as integrating policies and measures into cross-sector planning and decision-making processes. However, the implementation of such integration poses a challenge because of a lack of inter-and intra-institutional coordination combined with an inefficient multilevel governance context [59,80]. The current fragmented and multilayered governance arrangements with limited statuary powers need to be replaced with a new governance approach that can bring together the respected parties (including formal and informal actors) to undertake concerted actions [51,66,80,114].

According to Sharma & Singh [57], concerted actions in current urban governance processes are not effectively integrated across the sector or different levels of governance. There are also concerns that the machinery of policymaking is not flexible and malleable enough to face the uncertainties posed by climate change and disaster risk [65]. The current situation is that the judicial reviews of policy, ordinance, regulatory frameworks, and enforcement are not being revised continuously and embedded in the decision-making process [61,65,71]. Another barrier that has been identified under governance is short-term political mandates. Policies and strategies for CCA and DRR require long-term perspectives (including investment, process, commitment) that go beyond the normal duration of political mandates [52,56,116]. Political changes with different priorities tend to undermine a city's ability to achieve resilience and disrupt the long-term strategic vision.

Therefore, a new approach to governance that embraces greater collaboration, and that is flexible and adaptive, should be introduced to address the above shortcomings. The overall governance structures need to be refined using concepts, such as adaptive governance, to promote cross-organizational collaboration, power-sharing and public participation, transparency and accountability, and learning and adaptability [121,122]. Furthermore, the adaptive governance approach provides a generic and universal framework for policymakers to formulate policy when dealing with complex challenges, such as DRR and CCA [123].

*4.3. Fragmentation and Silo*

Table 9 shows issues and challenges under the fragmentation and silo theme.

**Table 9.** Issues and challenges under the fragmentation and silo theme.

| Issues and Challenges | Reference | Frequency (%) |
|---|---|---|
| Power relations | [49–51,55–59,61,66,68,80,86,87,90,94,106,109,110,113,115] | 3.8 |
| Weak coordination | [48,49,51,53,56–58,60,64,66,69,71–73,77,80,89,90,92,100,111] | 3.5 |
| Limited cooperation between stakeholders | [48,54,58,62,66,70,86,92,106,113–115,124,125] | 2.1 |
| Top-down approach | [50,60,61,63,76,86,90,94,106,115] | 1.9 |
| Lacks effective channel for interaction | [61,69,72,74,77,82,86,90,94,109] | 1.8 |
| No clear agreed vision | [48,49,54,59,61,62,72,74,78,81,116] | 1.8 |
| Trust matters | [63,68,92,93,111,113] | 0.9 |

It is evident that agencies involved in DRR, CCA, urban development, and the environment are working in silos and, therefore, are fragmented across several sectors [126]. Xu & Shao [90] highlight issues regarding diverse power relations between multiple agents in planning practices, both at the state and nonstate level. Although disaster management calls for shared governance between various levels [94], according to Sunarharum et al. [115], power is not shared in a bottom-up manner. Disaster risk governance is typically led by the federal government [94], which causes a top-down approach in decision-making. In such a scenario, the involvement of local governments is limited, with an inadequate distribution of legislative powers to the local governments. Such an approach has led to limited cooperation between the local and state levels, as well as between other stakeholders in the planning process [114]. Because of the disorganized and limited involvement of local actors during policy or strategy formulation, opportunities for cooperation between the relevant stakeholders have become limited [66,70,113,127]. Furthermore, the lack of effective channels for stakeholder interaction has introduced significant barriers to achieving cooperation since the input and needs of all the stakeholders are not adequately considered and actioned appropriately [72,82,90,94].

This research has found that the implementation of programs or activities for CCA and DRR are uncoordinated and are administered by a complex interplay between many different stakeholders acting in isolation [60,64]. Such a complicated interplay could lead to contradictions, overlapping roles and responsibilities, weak coordination and disorganized development [89]. Therefore, one of the main challenges in urban climate change adaptation is the coordination of the complex interplay between the stakeholders involved in the planning process in multiple institutional settings [60,89]. An effective coordination approach should have a clear distribution of responsibilities, authority, and the handling of resources. It is critical to consider power distribution in achieving effective coordination and, therefore, participatory approaches in CCA and DRR should be designed with the involvement of the state, local governments, and the community. However, a participatory approach could be seen as a threat at various levels of power-sharing and decision-making [113]. Hence, the involvement of stakeholders, with a clear understanding of their power, legitimacy, skills, experience, and knowledge, is required for the successful implementation of participatory approaches.

Other challenges identified in this theme include the lack of a clear agreed vision, and trust among the stakeholders. This lack of shared vision, missions, competencies, and goals between stakeholders can contribute to difficulties in planning [62]. Furthermore, mismatched perceptions among stakeholders have discouraged shared ownership and made developmental partners reluctant to work collaboratively for sustainable solutions and innovation [49]. Furthermore, a lack of trust between communities and authorities has been identified as a challenge due to the current bureaucracy and administrative issues that hinder healthy communication channels between the two parties [74,111].

To overcome the fragmentation and silos' issues in decision-making processes, an agile approach, suggested by Crawford et al. [128], can offer a potential solution for stakeholders to achieve a shared common vision and, thus, move towards successful outcomes. Adopted from the IT sector, the agile approach supports adaptive governance because of its flexibility in responding to changing circumstances [129]. From the perspective of project management, the agile approach provides stakeholders with opportunities for the codesign and coproduction of complex systems using user-centered design approaches, ensuring that the needs of the stakeholders are fully met [130]. The potential of the agile approach has already been demonstrated in the policy development process [131], and in the participatory planning process [132] using an agile participatory urban soundscape planning framework.

### 4.4. Capacity

Table 10 shows issues and challenges under the capacity theme.

**Table 10.** Issues and challenges under the capacity theme.

| Issues and Challenges | Reference | Frequency (%) |
|---|---|---|
| Inadequate skills and technical capacity | [48–50,52–54,56–59,61,64,66–70,72,80–83,86,92,96,110,112,115,116] | 5.7 |
| Lack of knowledge | [54,56,59,60,66,68,72,74,80,82,88,97,112,116] | 2.6 |
| Low adaptive capacity | [53,59,81,107,111–113] | 2.3 |
| Lack of awareness | [54,55,68,72,92,112,114] | 1.2 |
| Weak institutional leadership | [51,72,83,87,88,107,116] | 1.0 |

Ziervogel [133] defines urban transformative capacity as the extent of individuals' and organizations' ability to transform themselves and their society deliberately and consciously. This study shows that there is a lack of institutional capacity for handling climate change and adaptation at the local level. The main areas found to have a lack of capacity include inadequate skills and technical capacity in terms of information management [54,68,116], the lack of available expertise or qualified people in planning for CCA and DRR [56,64,72,92], and the lack of administrative and financial capacities [82].

Most local governments emphasize the difficulties of integrating CCA and DRR in the planning process because of a lack of knowledge to do so [60,116]. Such a lack of knowledge among authorities and decision-makers, combined with weak institutional leadership in planning and poor governance, can make cities vulnerable to a multitude of hazards [72,116] and climate change issues [53,81]. There are many examples where a poor understanding of climate science and vulnerability, and the lack of technical knowledge, has led to many ineffective development decisions [56,59,72].

In addition, the weak relationship that exists between local and national governments has resulted in very limited financial, technical, and human resources to implement CCA and DRR strategies at the local level [59,107,113]. According to Wamsler & Brink [111], the building up of citizens' adaptive capacities for climate change has been given less attention because of a lack of resources.

*4.5. Design & Development*

Table 11 shows issues and challenges under the design and development theme.

**Table 11.** Issues and challenges under the design and development theme.

| Issues and Challenges | Reference | Frequency (%) |
|---|---|---|
| Inadequate design consideration | [48,52,54,57,66,68,69,74–77,81,82,84,85,87,92,109,110,115,124,125] | 4.0 |
| Inappropriate development | [48,51,52,54,58,68,69,74,75,77,79,80,84,89,92,95,97,106,110] | 3.8 |
| Lack of legislative and regulatory integration in the planning process | [50,58,67–69,72,75,83,92,95,109,115] | 1.9 |
| Time requirement for implementation | [49,59,66,75,83,85,94,124] | 1.2 |
| Conventional design approach | [61,89,94,100,114] | 0.9 |
| Selection of construction material | [76,92,97,125] | 0.6 |

An adequate consideration of climate and disaster risk in the urban design process should assist in withstanding uncertainties in the future. A design process that involves potential users from an early stage will contribute to a thorough understanding of users, tasks, and environments, with a greater emphasis on a human-centered design approach [134]. Although the importance of considering urban disaster risks is known to avoid the catastrophic impacts of hazards, there is a lack of scientific analysis early in the design process with respect to risk assessment [124]. Moreover, the conventional design approach to resilience has been critiqued for not adequately considering the multiple stressors that shape urban societies [87].

The limited availability of land has led to inappropriate development within already vulnerable areas, consequently increasing further exposure to hazard [110]. Land resources become limited and unable to cater to rapidly expanding cities because of ineffective

land-use control [68] implemented by a fragmented and decentralised governance of metropolises [109]. In some cases, several irresponsible institutional decision-makers have allowed development in hazardous locations without considering the impact [54,89,92].

The integration of CCA and DRR in the planning process is constrained by different legislations and regulations, including policies, with most interventions undertaken in isolation [69,92,115]. It was also found that policies are not integrated into local planning regulations because of fragmented administrative systems with separated roles and responsibilities in urban planning, CCA, and DRR [58,72].

The implementation of CCA and DRR strategies will take time because of the longer planning horizon and development time required [49,83]. Factors, such as community disengagement or discouragement, and changes in budget administration, staff, priorities, and political direction can undermine the completion of proposed strategies if implementation has a long time frame [66,94]. Therefore, the management of this long-term process should consider time span as a key factor.

The consideration of appropriate building codes and minimum safety standards are not embedded in design parameters in many situations to reduce vulnerabilities for hazards [92]. The selection of appropriate construction materials is important to withstand the impact of disasters and reduce greenhouse gas emission targets and waste. For example, housing built using unsuitable construction materials will cause structures to become weak and increase the level of vulnerability [76,97,125]. Therefore, dealing with disasters and climate risk requires urban designers to understand the reason behind each design decision or consideration, as well as the time required for implementation and the materials or methods for construction.

*4.6. Data*

Table 12 shows issues and challenges under the data theme.

**Table 12.** Issues and challenges under the data theme.

| Issues and Challenges | Reference | Frequency (%) |
|---|---|---|
| Production of Information | [50–52,67,68,71–73,75,77,80,81,83,85,92, 96,97,107,108,111,114–116,124,125,135] | 6.3 |
| Data sharing | [68,73,92,95,114,116,135] | 1.2 |
| Lack of access to information and verification | [50,51,92,96,113,116] | 1.2 |
| Lack of critical local details | [55,63,75,77,88,89] | 0.9 |
| Complex data sets | [68,83] | 0.3 |

Previous scholars, such as Hassani et al. [136], and Mall et al. [137], have highlighted how numerous data contribute to the disaster and climate change field. Hardoy et al. [116] highlight the gap between the production of information and its use for decision-making. Filho et al. [107] argue that the use of unreliable data for prioritizing adaptation actions, the limited availability of data to understand risk, and the difficulties in integrating scientific information into legislation, hamper the effectiveness of decision-making [107]. Furthermore, incomplete baseline information on factors relating to disasters and climate change will impede any efforts made to resolve urban development issues. For example, Sunarharum et al. [115] report that their GIS application for decision-making could not be fully utilized because of the absence of complete data. The lack of information (such as on-time series data on cities for climate change projection, urban population, exposure, and vulnerability) undermines effective risk evaluation, evidence-based policymaking, and effective planning and development [51,67,108].

Therefore, the availability and accessibility of information are crucial for developing climate action plans and planning and building to withstand disaster risks [50,51]. However, access to appropriate information is an enormous challenge since the information is usually held by different actors in isolation [50,116]. In this context, data-sharing between

institutions or agencies is often difficult and frequently incompatible because of the lack of shared terminology and techniques [95,116] and data sensitivity [135]. Although it is recognized that critical local details on communities (such as the profile of the community, and the needs and common issues affecting community livelihoods) provide a foundation in decision-making, it has been reported that regional planning and strategy development activities at the local level sometimes lack critical local details [63,89]. Consequently, community concerns are not adequately addressed [55]. In addition, the use of inappropriate methods in presenting complex datasets will hinder stakeholder engagement since they will not be able to interpret them and provide a meaningful contribution to the discussion. Therefore, the datasets need to be presented in a simple form and made accessible for users to support their decision-making [68,83].

The above challenges can be overcome by developing a digital platform that can combine a multitude of information from various sources as the basis for decision-making processes, as pointed out in the literature by Van Westen [138]. Furthermore, related studies have shown that the presentation of hazards, vulnerability, coping capacities and risks in the form of digital maps offers a highly effective, easy-to-use visual communication medium [139–141], allowing users to become active participants [142], hence promoting participatory planning.

### 4.7. Funding

Table 13 shows issues and challenges under the funding theme.

**Table 13.** Issues and challenges under the funding theme.

| Issues and Challenges | Reference | Frequency (%) |
|---|---|---|
| Unequal distribution of funds | [51–54,56,57,61,68,70,74,80,81,85,90] | 2.6 |
| Lack of implementation funding | [17,51,54,58,59,64,70,81,86,92] | 1.6 |
| Absence of financial support at the local level | [52,59,64,66,85,107] | 1.0 |
| Bureaucracy | [50,64,69,80,82,94,111] | 1.0 |
| Economic | [49,52,60,64,69] | 1.0 |

Implementing climate actions and disaster risk reduction is impossible without appropriate funding mechanisms [51,81,85]. The lack of dedicated funding at the national level for risk reduction, with priorities being given to relief, recovery, and rehabilitation by governments, has been identified as a major issue [53,68]. At the local level, the absence of financial support limits the capacity of local governments to address CCA and DRR issues and, hence, the reliance on the private sector for implementation [52]. Because of this unequal distribution of funds for implementing climate and disaster risk protection measures, insufficient attention is given to addressing community needs [57,61,74]. Therefore, there is a need to consider new funding mechanisms to address this unequal distribution of funds to support cross-cutting issues, such as disasters and climate change [54]. These new funding mechanisms should also address issues such as complex bureaucratic procedures and political influence in securing government funding [80,111]. From the economic perspective, one important consideration is to have a cost-benefit analysis of adaptation measures that provides evidence on gains and losses to convince decision-makers [81]. Thus, there is a need for establishing new funding mechanisms with long-term plans to provide equitable and sustainable financial assistance in building resilience for cities.

Although global agendas, such as the Sendai Framework (priority 3), advise governments to invest in disaster risk reduction for resilience, there is little evidence to show that this is fully implemented. However, it is hoped that the impetus brought about by the Leader Summit on Climate Change in April 2021 will help to address the national governments' funding deficiency in considering DRR and CCA in urban development processes.

## 5. Conclusions and Future Research Directions

This study was motivated by the challenge of shifting current urban planning practices into a new innovative approach that can enable a transition towards risk-sensitive urban development across all scales and contexts. The systematic review conducted in this study identified forty-six issues and challenges under seven key themes (trade-offs, governance, fragmentation and silos, capacity, design and development, data and funding) that need addressing in order to facilitate the desirable transition.

This research found that conflicts among stakeholders exist because of the differences in public and political priorities, short-term and long-term goals, temporal and spatial goals, urban growth and ecology concerns, and equity and equality needs. These conflicts need to be addressed through open and transparent trade-off discussions through participatory planning that challenges existing systems, processes, social values, institutions, and technical practices and paradigms. However, the implementation of a participatory approach for negotiating trade-offs needs to be supported with an evidence-based approach based on the availability of information collected from trusted sources and scientific simulation models to ensure sound decisions in disaster risk reduction and climate change adaptation [107,116,136,137]. Not having a sound evidence-based decision-making process could otherwise lead to the narratives presented by actors with power and authority being dominant in the trade-off negotiation processes. This research shows the limited availability of comprehensive datasets for stakeholders to understand hazards, vulnerability, exposure, and climate change impacts. Additionally, there is a lack of data-sharing practices and difficulty in integrating scientific information in the decision-making process to analyze various conflicting narratives and to choose the most appropriate solutions that are equitable. Hence, there is a need for introducing digital solutions that promote data collection, data-sharing practices, and data-driven decision-making processes among stakeholders.

This research shows that many challenges exist within the context of the overall governance of risk-sensitive urban development (including weak governance; deficiencies in equity policies; weak institutional coordination; unclear separation of roles and responsibilities; complex governance structures and arrangements; a lack of equity policies; poor regulatory instruments; a lack of mainstreaming in DRR and CCA; ineffective integration in urban government systems, and short-term political mandates). These challenges need to be addressed by introducing a new approach to governance, such as adaptive governance [24,121,122], that promotes cross-organizational collaboration, power-sharing and public participation, transparency and accountability, and learning and adaptability. In addition to governance, fragmentation and the silo-based practices that exist among government organizations were identified as a major area of concern in this study. The major issues and challenges found under fragmented and silo-based practices are the weak coordination and cooperation between state and local organizations; power relations; and a lack of trust, communication, and a clear collective vision. The agile approach, typically used in the IT sector, has the potential to address these challenges, allowing stakeholders to achieve a shared common vision, and to codesign and coproduce potential solutions.

This research found that there are many limitations in current urban design and development processes (including inadequate scientific analysis during the design process, a lack of policy implementation in planning and development that prohibits expansion in hazardous locations, inefficient land-use control, and the use of unsuitable construction materials). The integration of CCA and DRR in the planning process is currently constrained by fragmented administrative systems, different legislations, regulations, and policies. Furthermore, this study shows that there is a lack of institutional capacity and there are deficiencies in skills, knowledge, awareness, and leadership for handling climate change and adaptation at the local level. In this aspect, a social learning process is proposed to encourage mutual learning and to improve collective decision-making. This approach enables stakeholders to understand the multiple perspectives of complex problems and will stimulate progress in building capacity through active involvement and experimentation.

The major obstacle to addressing the identified challenges in this study is the availability of funding from national governments. This research identified the key barriers under funding as: the focus of national governments on relief, recovery, and rehabilitation rather than mitigation and adaptation for DRR and CCA; a lack of support at the local level; and a lack of understanding of the long-term cost benefits. Hence, it is important to address the current unequal distribution of funds to support cross-cutting issues, such as disasters and climate change.

In summary, this study has conducted a comprehensive analysis of the major challenges that hinder the implementation of risk-sensitive urban development that is paramount in creating resilient cities against climate change. However, it is important to note that the findings of this study are limited to articles found by the search query and selected databases and, therefore, several concepts that relate to this topic might not be covered or explained in detail. It is hoped that this paper will ignite further research that will seek solutions to overcome the challenges identified under the themes of trade-offs, governance, fragmentation and silos, capacity, design and development, data and funding. Some examples of research questions that need further investigation are: How can the participatory processes be conducted in a fair, transparent, and scientifically sound manner?; How does a community handle power, advocacy, equity, justice, ethics, and knowledge exploitation in participatory planning?; What narratives are being transmitted, and who is communicating various narratives and why?; What narratives need to be developed, presented, and discussed to establish a comprehensive understanding of the impact of the proposed developments on the community, economy, and environment?; How can complex risk trade-off narratives be communicated to nonexperts to understand and build consensus?; What are the conflicting narratives and the types of trade-offs that need to be considered in promoting a new form of development practice that is equitable and resilient and considers vulnerable communities, DRR, and CCA as the key factors?; What formal and informal structures can facilitate the implementation of adaptive governance? How can we adopt agile methodology for bringing organizations to implement risk-sensitive thinking in urban development projects?; What is the nature of a digital platform that can support a collaborative approach to evidence-based decision-making among stakeholders?

**Author Contributions:** Conceptualization: T.F., S.B., N.S. and A.F.R.; methodology: A.F.R.; validation T.F., S.B. and N.S.; writing—original draft preparation: A.F.R., T.F.; writing—review and editing: T.F., S.B., N.S. and A.F.R. All authors have read and agreed to the published version of the manuscript.

**Funding:** This work was supported by the Global Challenges Research Fund (GCRF) and the Economic and Social Research Council (ESRC) under the Grant ES/T003219/1 entitled "Technology Enhanced Stakeholder Collaboration for Supporting Risk-Sensitive Sustainable Urban Development."

**Data Availability Statement:** Data-sharing does not apply to this article.

**Acknowledgments:** The authors would like to acknowledge the Construction Research Institute of Malaysia (CREAM), especially Zuhairi Abd. Hamid for support. The authors also want to acknowledge Hanneke and the TRANSCEND project team members for their insightful feedback.

**Conflicts of Interest:** The authors declare no conflict of interest.

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
