# Peer review of "Transformation towards Risk-Sensitive Urban Development: A Systematic Review of the Issues and Challenges"

_sustainability, doi:10.3390/su131910631_

Round 1
Reviewer 1 Report
Summary
The study conducts a systematic review based on the PRISMA method to detect issues and challenges in implementing risk-sensitive urban development. The analysis is well designed, with logical coherence. The research question is clearly identified, but of no great novelty. Extracted data and results are interesting and presented accurately. The text is clearly organized and fluent.
Comments
It is suggested to revise accurately the following text lines:
- Lines 24–25. Keywords do not contain crucial terms: e.g., disaster risk reduction, climate change adaptation, decision-making processes, and collaborative planning approaches.
- Line 37. Relevant stakeholders in implementing risk-sensitive urban development are not clearly identified.
- Line 38. A proper definition of urban risks, underpinned by key literature, is not provided.
- Lines 174–75. In Table 4, the justification column is not particularly explanatory and supplementary to the criteria and question columns.
- Line 177. It is not explicit which arguments needed to be supported integrating further 14 papers.
- Line 204. It would be useful to understand which hazards and risks were addressed in the assessed articles (if applicable). It would put in focus contextual problems/issues. Moreover, it would provide readers with a clear overview of selected contents for assessment.
- Line 604. The sentence “in a new visual way visually” sounds not appropriate.
More broadly, the implication of this study for future research (e.g., through additional investigation) appears not exhaustively discussed.
Reviewer 2 Report
This work deals with a topic of high interest for the scientific community since the debate on how to mainstream the risk assessment into planning is still an uncovered issue. Overall, this manuscript has a sound structure and employ bibliographic research that adopts certain criteria to select some papers and examine their main contribution to this field of research. 7 key issues are used to discuss the results. To what concern the contents the main problem of this manuscript is the absence of an adequate problematization of the context of risk (IPCC) and how this concept is related to resilience (that is not sustainability), vulnerability, sensitivity, hazard and coping capacities. All these terms are present here and there and constitute an abundant part of the debate but the authors did not introduce their meaning in the first part of the work. Therefore, some results are difficult to understand since the premises of these "interpretations" are not clarified. For example, risk management is not mainstreamed because many people still confuse vulnerability with risk or resilience, even in the GIS diagnosis, and this constitutes a big weakness. The method part seems to be clear. Certain point should be clarified (see my attached file). Discussion, on the other hand, is long, often repetitive and overlapping (are you sure that you need 7 issues?) and, besides, is ofter vague since is composed of general statements extracted from bibliography without ad empirical context-based evidence. I suggest here to reduce and synthesize while completing this part with a clear policy suggestion statement that concludes this work.
Detailed comments are in the attached file.
Good Luck!

Reviewer 3 Report
Dear Authors,
I read the article with great interest. Regardless of the interesting general problem, its topicality is noteworthy. Its overview character is also essential.
The article is written very well when it comes to the substantive scope. The authors are well-read and know the literature on the subject well.
My most significant doubts are the layout of the text and specified the proportions of individual chapters.
Apart from the Introduction, the other subsections are written in an overly individual layout. In my opinion, this causes disproportions between what the Authors want to convey to the Readers and what the Readers of Sustainability expect.
Therefore, I propose to change the layout of the article as follows:
- Introduction, including the purpose of the research and why this article is essential, what it brings new compared to the previous ones.
- Review of research to date. Who has done such systematization before, what has succeeded and what has not? Maybe they were similar or covered a smaller range of problems?
- Description of data and methods.
- Results.
- Discussion. There should be no elements typical for Results (I see them in the current version of the article). The authors should discuss here the scientific achievements to date. Identify the strengths and weaknesses of your research, present challenges).
- Conclusion.
Authors should also take care of the appropriate length of individual chapters of the article. Discussion and Conclusion are too long in this version. In my opinion, some issues that are in these chapters should be in the Results section.
Reviewer 4 Report
This paper is really a literature review about disaster risk resiliency that doesn't really connect to the SDGs or the UN framework that is briefly mentioned in the introduction. While the topic itself is interesting, the presentation of the classification scheme would be useful for quickly tracking sources in a spreadsheet, but not for understanding the differences between the authors in this field. Data presented in the tables is not really integrated in the text around the tables.
The conclusion is not supported in the text, it appears to be a summary of other authors work rather than a conclusion based on the methodology used to establish the literature review.
There is no discussion of the frequency data presented. A random search of references shows that the references some references only appear in the tables and not in the text.
Specific comments are presented by line number below.
There are two run on sentences at line 38 and again at line 55.
Line 144. Why did you use 2010 as the cutoff year? This would include articles pre-SDGs and creates a conflict between your methodology and your framing.
Line 177. There is no justification presented for adding an additional 14 papers to this study. Given that this addition is a significant expansion to the number of papers reviewed, there should be a justification stated.
Line 181. There is a mismatch between section 3, the introduction, and the abstract. This paper is really a literature review that is not connected to the creation or implementation of the SDGs. I would recommend removing the SDG phrasing and leaving this as a literature review.
Line 208. I find this paragraph about which journals publish articles on disaster resilience/climate change to be irrelevant. There is nothing new or unexpected about journals in the disaster resilience and/or the built environment field publishing papers on disaster resilience and/or the built environment.
Line 296 Two undefined acronyms are used in this line.
Round 2
Reviewer 2 Report
I think now this work gained in quality and clarity. The authors made a diffuse deep revision of the manuscript while taking into account all the considerations made in the first round. Therefore I don't have any further (structural) suggestions. Only re-read carefully avoiding some minor mistakes (see the attached file) and try to avoid (where possible) some repetitive/vague statements.
Good luck!

Reviewer 3 Report
Dear Authors,
I welcomed the significant improvement of the article by the Authors. I believe that the Methods chapter could be slightly modified. Some information from the Methods chapter should be included in the Results chapter.
Reviewer 4 Report
While the authors have addressed the concerns from the original review, the paper is significantly improved from the previous version submitted. However, for me the suitability of the paper for publication remains in doubt as I cannot identify what the original contribution to the field will be given this is a research project whose case study involves previously published papers.
I also note that the results of this study are not replicable. Fourteen of the 82 papers assessed were included as a judgement call made by the authors (see in lines 189 - 190) without any type of rationale other than they related to "particular topics or approaches discovered...."
Round 3
Reviewer 4 Report
The issues I am concerned about have been resolved.